# Investigating dosage effects of ovulation inhibitors on oocyte maturation in assisted reproductive technology: A retrospective study among patients with normal ovarian reserve

Mika Handa[1☉], Tsuyoshi Takiuchi[1,2☉*], Sumika Kawaguchi[3], Yasuhiro Ohara[1,4], Masakazu Doshida[5], Takumi Takeuchi[5], Hidehiko Matsubayashi[4,5], Tomomoto Ishikawa[4,5], Sho Komukai[6], Tetsuhisa Kitamura[7], Tadashi Kimura[1]

1 Department of Obstetrics and Gynecology, Osaka University Graduate School of Medicine, Suita, Osaka, Japan, 2 Department of Clinical Genomics, Osaka University Graduate School of Medicine, Suita, Osaka, Japan, 3 Clinical Study Support Center, Wakayama Medical University Hospital, Wakayama, Wakayama, Japan, 4 Department of Reproductive Medicine, Reproduction Clinic Osaka, Osaka, Osaka, Japan, 5 Department of Reproductive Medicine, Reproduction Clinic Tokyo, Minato-Ku, Tokyo, Japan, 6 Division of Biomedical Statistics, Department of Integrated Medicine, Osaka University Graduate School of Medicine, Suita, Osaka, Japan, 7 Division of Environmental Medicine and Population Services, Department of Social and Environmental Medicine, Osaka University Graduate School of Medicine, Suita, Osaka, Japan

☉ These authors contributed equally to this work.
* takiuchi.tsuyoshi.med@osaka-u.ac.jp

**Data Availability Statement:** The data cannot be shared publicly due to the presence of potentially

## Abstract

The judicious selection of ovulation inhibitors in ovarian stimulation protocols is crucial for the success of assisted reproductive technology (ART). Herein, we investigate the dose-dependent effects of chlormadinone acetate (CMA) and cetrorelix, two distinct ovulation inhibitors, on oocyte maturation in patients with normal ovarian reserve, using univariable and multivariable Poisson regression analyses. Patients undergoing progestin-primed ovarian stimulation (PPOS) with CMA ($n = 299$) or gonadotropin-releasing hormone antagonist (GnRH-ant) with cetrorelix ($n = 605$) during their initial *in vitro* fertilization cycle were enrolled at our center from March 2018 to October 2020 ($N = 904$). The primary and secondary outcomes were the oocyte maturation and fertilization rates, respectively. After adjusting for several covariates including age, anti-Müllerian hormone levels, total gonadotropin dose, and type of trigger, we calculated the dose-dependent adjusted relative risk (aRR) and 95% confidence interval (CI) for 1 mg of CMA or 0.25 mg of cetrorelix. In the PPOS group, the median age was 34.0 years, and the median total CMA dosage was 22 mg (interquartile range [IQR]: 18.0–32.0). In the GnRH-ant group, the median age was 35.0 years, and the median total cetrorelix dosage was 0.5 mg (IQR 0.5–0.5). The aRR of the maturation rate was 1.003 (95% CI: 0.999–1.007) with PPOS ($p = 0.194$) and 1.009 (95% CI: 0.962–1.059) with GnRH-ant ($p = 0.717$). The aRR of the fertilization rate was 1.002 (95% CI: 0.985–1.020) with PPOS ($p = 0.783$) and 1.022 (95% CI: 0.839–1.246) with GnRH-ant ($p = 0.829$). Collectively, these findings indicate that within the applied dosages, ovulation inhibitors do not significantly impact oocyte maturation or fertilization rates in patients with normal ovarian

identifying or sensitive patient information. However, researchers who meet the criteria for access to confidential data may obtain the data from the steering committee of the study, specifically the Ethical Review Board of Osaka University Hospital (contact: rinri@hp-crc.med. osaka-u.ac.jp).

**Funding:** This work was supported by JSPS (Japan Society for the Promotion of Science) KAKENHI (Grant-in-Aid for Scientific Research (C)) to M.H. [grant number 23K08867]. JSPS website; https:// www.jsps.go.jp/english/ The funders had no role in study design, data collection and analysis, decision to publish, or preparation of the manuscript.

**Competing interests:** The authors have declared that no competing interests exist.

reserve. These valuable insights can be applied when designing ART protocols and may guide clinicians in optimizing infertility treatments.

## Introduction

Globally, approximately 8–12% of reproductive-age couples grapple with infertility, particularly in developed countries, further intensifying the declining birth rates [1]. Infertility extends beyond mere health issues, inflicting substantial psychological and economic stress and exacerbating psychosocial distress in affected females [2]. Hence, the implications transcend individual health concerns, posing substantial challenges to societal demographics and family structural dynamics. With over three million cycles conducted in 2018, assisted reproductive technology (ART) has been progressively utilized worldwide [3]. However, the rate of successful live births remains comparatively modest [3]. This underscores the urgent need for continuous research and improvements in ART methodologies and practices. ART primarily comprises four processes: oocyte retrieval following ovarian stimulation, fertilization of oocytes with sperm, culturing of the resulting embryos, and embryo transfer [4]. The retrieval of a sufficient number of high-quality oocytes is a crucial aspect of ART. Therefore, selecting and understanding appropriate controlled ovarian stimulation (COS) methods are of paramount importance.

COS methods largely comprise the administration of ovulation inhibitors to prevent luteinizing hormone (LH) surge and subsequent ovulation of developed oocytes before oocyte retrieval. First, an agonist method, initially using gonadotropin-releasing hormone (GnRH) agonists as ovulation inhibitors, was developed [5]. This was followed by the widespread adoption of the GnRH antagonist (GnRH-ant) protocol, involving the administration of GnRH antagonists as ovulation inhibitors [6] and, more recently, the progestin-primed ovarian stimulation (PPOS) protocol, which uses progestins [7, 8].

Although ovulation inhibitors effectively suppress ovulation, they concurrently inhibit the secretion of LH and follicle-stimulating hormone (FSH), which are essential for oocyte development [9]. Consequently, the excessive use of these inhibitors may hinder oocyte maturation and development, ultimately impeding the retrieval of good-quality oocytes. Although previous studies have compared the impact of ovarian stimulation methods that use different daily ovulation inhibitor dosages on the number of oocytes and metaphase II (MII) oocytes. [7, 10, 11], they have not addressed the clinical question of which ovulation inhibitor doses within each method can be safely administered without compromising oocyte quality. Indeed, higher doses of ovulation inhibitors may negatively affect both the maturation and developmental potential of oocytes. To clarify this, the dose-dependent impact of total ovulation inhibitor use on oocyte quality must be assessed after adjusting for various confounding factors.

Accordingly, in the current study, we employ Poisson regression analysis to assess the dose-dependent effects of ovulation inhibitors on oocyte maturation and fertilization rates using the PPOS and GnRH-ant protocols. Our findings demonstrate that within the applied dosages, these ovulation inhibitors did not significantly affect oocyte maturation or fertilization rates, providing valuable insights for optimizing infertility treatments and guiding the design of ART protocols.

## Materials and methods

### Study population

This retrospective cohort study, conducted at a reproduction center between March 2018 and October 2020, included 1,298 Japanese patients with normal ovarian reserve undergoing their

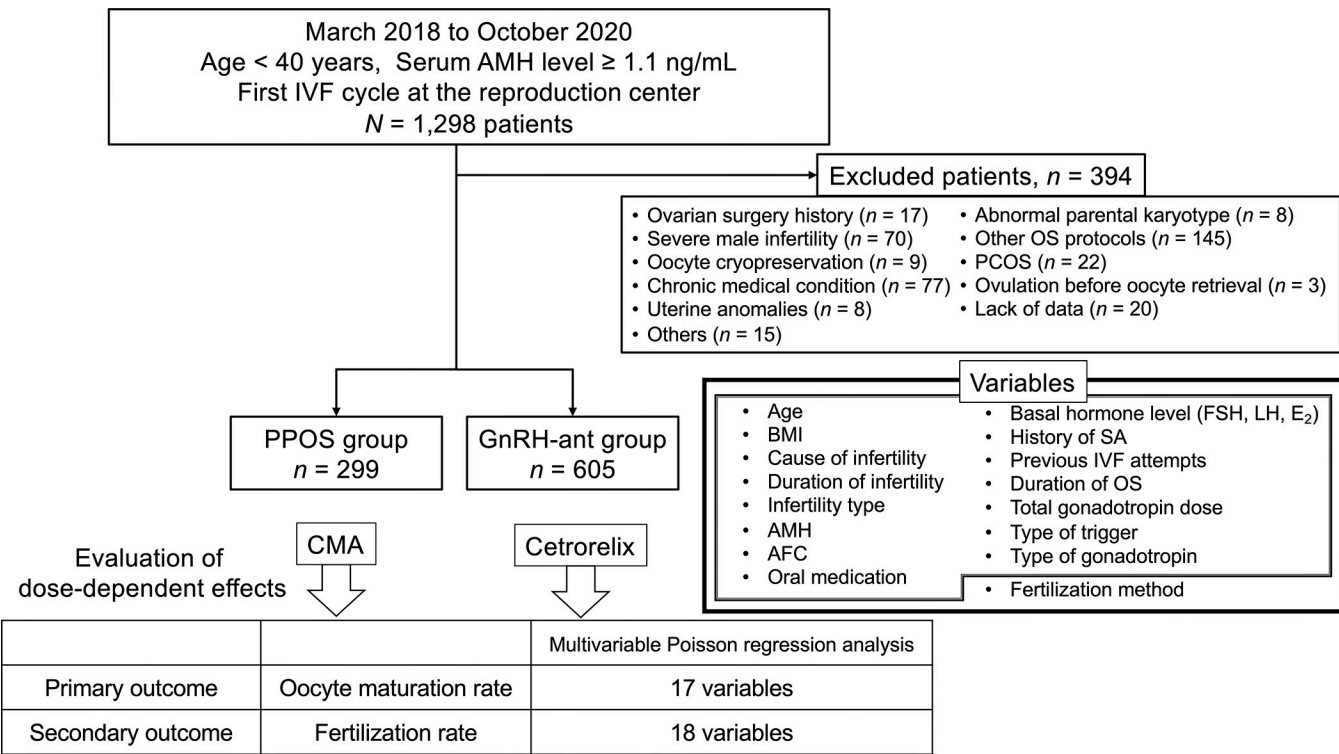

**Fig 1. Flowchart of the study population selection and Poisson regression analysis.** A schematic representation illustrating the flow of participant selection and the subsequent Poisson regression analysis in a retrospective study conducted at a reproduction center. The study involved 1,298 Japanese patients with normal ovarian reserves undergoing their initial in vitro fertilization (IVF) cycles, categorized into two groups: progestin-primed ovarian stimulation (PPOS) with chlormadinone acetate (CMA) and GnRH-antagonist (GnRH-ant) with cetrorelix. AMH, anti-Müllerian hormone; BMI, body mass index; CMA, chlormadinone acetate; $E_2$, estradiol; FSH, follicle-stimulating hormone; GnRH-ant, GnRH antagonist; IVF, in vitro fertilization; LH, luteinizing hormone; OS, ovarian stimulation; PCOS, polycystic ovary syndrome; PPOS, progestin-primed ovarian stimulation; SA, spontaneous abortion. Basal hormone levels refer to serum concentrations of FSH, LH, and $E_2$.

first *in vitro* fertilization (IVF) cycles. Participants were divided into two groups: PPOS with chlormadinone acetate (CMA) and GnRH-ant with cetrorelix. The inclusion criteria comprised individuals < 40 years of age, an anti-Müllerian hormone (AMH) level ≥ 1.1 ng/mL, and patients who had their own oocytes retrieved in their initial IVF cycle at our center. These criteria aligned with the lower limits established by the Bologna criteria for poor responders, characterized by a suboptimal response to COS, resulting in diminished oocyte retrieval and fewer embryos for transfer [12]. Patients with a history of ovarian surgery, oocyte cryopreservation, chronic diseases (such as cancer or diabetes), congenital uterine anomalies, polycystic ovary syndrome, or couples with severe forms of male infertility necessitating simple or microdissection testicular sperm extraction or chromosomal abnormalities, and those with incomplete or missing data were excluded. Additionally, patients who exhibited signs of ovulation before oocyte retrieval, which were defined as a serum progesterone level ≥ 5.0 ng/mL or ultrasonographical evidence of ovulation (disappearance of the primary follicle, corpus luteum formation, and appearance of ascites fluid), were excluded. Early ovulation before oocyte retrieval did not occur in the PPOS group, whereas it occurred in three patients in the GnRH-ant group, who were subsequently excluded from the analysis. After excluding 394 patients, 904 were included, with 299 undergoing PPOS with CMA and 605 undergoing GnRH-ant with cetrorelix (Fig 1).

### Treatment protocol

**Controlled ovarian stimulation (COS).** The baseline characteristics of patients undergoing COS procedures are detailed in Table 1, presenting median and interquartile ranges (IQR). COS was initiated between Days 2 and 5 of the menstrual cycle. The initial dose of human menopausal gonadotropin (hMG; HMG Ferring; Ferring Pharmaceuticals, Tokyo, Japan; HMG Fuji; Fuji Pharma, Tokyo, Japan) or rFSH (Gonal-f; Merck, Tokyo, Japan) ranged from 150 to 450 IU, considering age, AMH levels, and body mass index (BMI). In the PPOS protocol, 2 mg of CMA (Lutral tablets; Fuji Pharma) was orally administered daily from Day 2 to Day 5 of the menstrual cycle until the trigger day. When the serum LH level exceeded 5 mIU/mL, the CMA dose was increased by 2 mg/day (up to a maximum of 6 mg).

In the GnRH-ant protocol, patients received cetrorelix (0.25 mg/mL; Merck, Tokyo, Japan) every other day, commencing between Day 8 and Day 10 of the cycle or when leading follicles reached $\geq 14$ mm in diameter. The COS protocol is shown in Fig 2. Patients at high risk of ovarian hyperstimulation syndrome, identified by a serum AMH level $\geq 5.0$ ng/mL or antral follicle count (AFC) $\geq 15$ on transvaginal ultrasonography, were administered 2.5 mg/day of an aromatase inhibitor (Femara; Novartis, Tokyo, Japan) for 2–5 days. Those with a serum AMH level $< 2.0$ ng/mL received 50–100 mg of clomifene citrate (Clomid; Fuji Pharma) daily throughout stimulation. Ovulation was induced with human chorionic gonadotropin (hCG; HCG Mochida; Mochida Pharmaceutical Co., Tokyo, Japan), a GnRH agonist (Buserecur; Fuji Pharma), or a dual trigger when leading follicles exceeded $> 18$ mm in diameter, followed by oocyte retrieval $36 \pm 2$ h later. Oocytes were fertilized via conventional IVF, intra-cytoplasmic sperm injection (ICSI), or split-ICSI, based on semen parameters and the number of retrieved oocytes.

**Outcomes.** The primary outcome was the oocyte maturation rate. For both ICSI and cIVF methods, the oocyte maturation rate was calculated as the number of mature oocytes divided by the total number of oocytes retrieved. In ICSI, mature oocytes were specifically defined as those in the MII stage. In cIVF method, mature oocytes were identified by the presence of a visible polar body, observed the day after insemination, with germinal vesicle and metaphase I stage oocytes excluded. The secondary outcome was the fertilization rate, defined as the ratio of two pronuclear embryos to the number of retrieved oocytes with cIVF or mature oocytes with ICSI.

### Statistical analysis

The primary outcome was analyzed using univariable and multivariable Poisson regression analyses, incorporating 17 variables: age, BMI, cause of infertility, duration of infertility, infertility type, serum AMH level, basal hormone level (FSH, LH, and estradiol [$E_2$]), antral follicle count, history of spontaneous abortion, previous IVF attempts, duration of ovarian stimulation, total gonadotropin dose, type of oral medication, type of gonadotropin, and type of trigger. These covariates were selected a priori based on clinical plausibility and previous studies [13, 14]. The secondary outcome was subjected to underwent both univariable and multivariable Poisson regression analyses with adjustments for 18 variables, including the previously mentioned 17 factors and fertility methods. The unadjusted relative risk (RR) and adjusted RR (aRR) were calculated, and the 95% confidence interval (CI) per 1 mg of CMA or 0.25 mg of cetrorelix increased in a dose-dependent manner. All tests were two-tailed, with $p$-values $< 0.05$ considered statistically significant for comparisons between the two groups. R software version 4.2.2 R was used for all statistical analyses of clinical data.

**Table 1. Baseline characteristics of patients undergoing the progestin-primed ovarian stimulation (PPOS) or GnRH-antagonist (GnRH-ant) protocols.**

| | | | PPOS | GnRH-ant |
|---|---|---|---|---|
| | | | *n* = 299 | *n* = 605 |
| **Age at oocyte retrieval (median [IQR])** | | | 34.0 [32.0–37.0] | 35.0 [32.0–37.0] |
| **BMI (median [IQR])** | | | 20.4 [19.1–22.1] | 20.3 [19.1–21.9] |
| **Cause of infertility (*n* (%))** | | Tubal factor | 19 (6.4) | 26 (4.3) |
| | | Male factor | 97 (32.4) | 164 (27.1) |
| | | Endometriosis | 10 (3.3) | 26 (4.3) |
| | | Others | 173 (57.9) | 389 (64.3) |
| **Duration of infertility (months) (median [IQR])** | | | 24.0 [14.0–41.0] | 24.0 [15.0–40.0] |
| **Type of infertility (*n* (%))** | | Primary | 200 (66.9) | 395 (65.3) |
| | | Secondary | 99 (33.1) | 210 (34.7) |
| **Serum AMH level (ng/mL) (median [IQR])** | | | 3.90 [2.50–5.67] | 3.55 [2.20–5.78] |
| **Basal hormone level (median [IQR])** | | FSH (mIU/mL) | 8.3 [7.2–9.8] | 8.7 [7.5–10.0] |
| | | LH (mIU/mL) | 4.8 [3.7–6.2] | 4.2 [3.0–5.5] |
| | | $E_2$ (pg/mL) | 33.9 [26.1–45.0] | 32.8 [24.2–44.5] |
| **AFC (median [IQR])** | | | 11.0 [7.0–15.0] | 10.0 [7.0–15.0] |
| **History of SA (*n* (%))** | | 0 | 245 (81.9) | 476 (78.7) |
| | | 1 | 44 (14.7) | 95 (15.7) |
| | | 2 | 9 (3.0) | 23 (3.8) |
| | | 3 | 1 (0.3) | 9 (1.5) |
| | | 4 | 0 (0) | 1 (0.2) |
| | | 5 | 0 (0) | 0 (0) |
| | | 6 | 0 (0) | 1 (0.2) |
| **Previous IVF attempts (*n* (%))** | | 0 | 163 (54.5) | 243 (40.2) |
| | | 1–2 | 81 (27.1) | 213 (35.2) |
| | | $\geq 3$ | 55 (18.4) | 149 (24.6) |
| **Duration of OS (days) (median [IQR])** | | | 9.0 [8.00–10.0] | 9.0 [8.00–10.0] |
| **Total gonadotropin dose (IU) (median [IQR])** | | | 2400 [1613–2700] | 2400 [1800–2700] |
| **Gonadotropin type (*n* (%))** | | hMG only | 141 (47.2) | 299 (49.4) |
| | | hMG+rFSH | 6 (2.0) | 7 (1.2) |
| | | rFSH only | 152 (50.8) | 299 (49.4) |
| **Oral medications administered for ovarian stimulation (n (%))** | | None | 101 (33.8) | 173 (28.6) |
| | | Clomiphene citrate | 105 (35.1) | 242 (40.0) |
| | | Aromatase inhibitor | 93 (31.1) | 190 (31.4) |
| **Type of trigger (*n* (%))** | | GnRH agonist | 40 (13.4) | 30 (5.0) |
| | | hCG 3000 IU | 0 (0) | 4 (0.7) |
| | | hCG 5000 IU | 48 (16.1) | 112 (18.5) |
| | | hCG 10000 IU | 113 (37.8) | 187 (30.9) |
| | | hCG + GnRH agonist | 98 (32.8) | 272 (45.0) |
| **Fertilization method (*n* (%))** | | cIVF | 33 (11.0) | 91 (15.0) |
| | | ICSI | 93 (31.1) | 155 (25.6) |
| | | Split-ICSI | 173 (57.9) | 359 (59.3) |

This table provides a detailed overview of the baseline characteristics of patients undergoing controlled ovarian stimulation (COS) procedures. AFC, antral follicle count; AMH, anti-Müllerian hormone; BMI, body mass index; $E_2$, estradiol; FSH, follicle-stimulating hormone; GnRH-ant, GnRH-antagonist; hCG, human chorionic gonadotropin; hMG, human menopausal gonadotropin; ICSI, intra-cytoplasmic sperm injection; IQR, interquartile range; IVF, *in vitro* fertilization; LH, luteinizing hormone; OS, ovarian stimulation; PPOS, progestin-primed ovarian stimulation; rFSH, recombinant human follicle-stimulating hormone; SA, spontaneous abortion; basal hormone levels refer to serum concentrations of FSH, LH, and $E_2$.

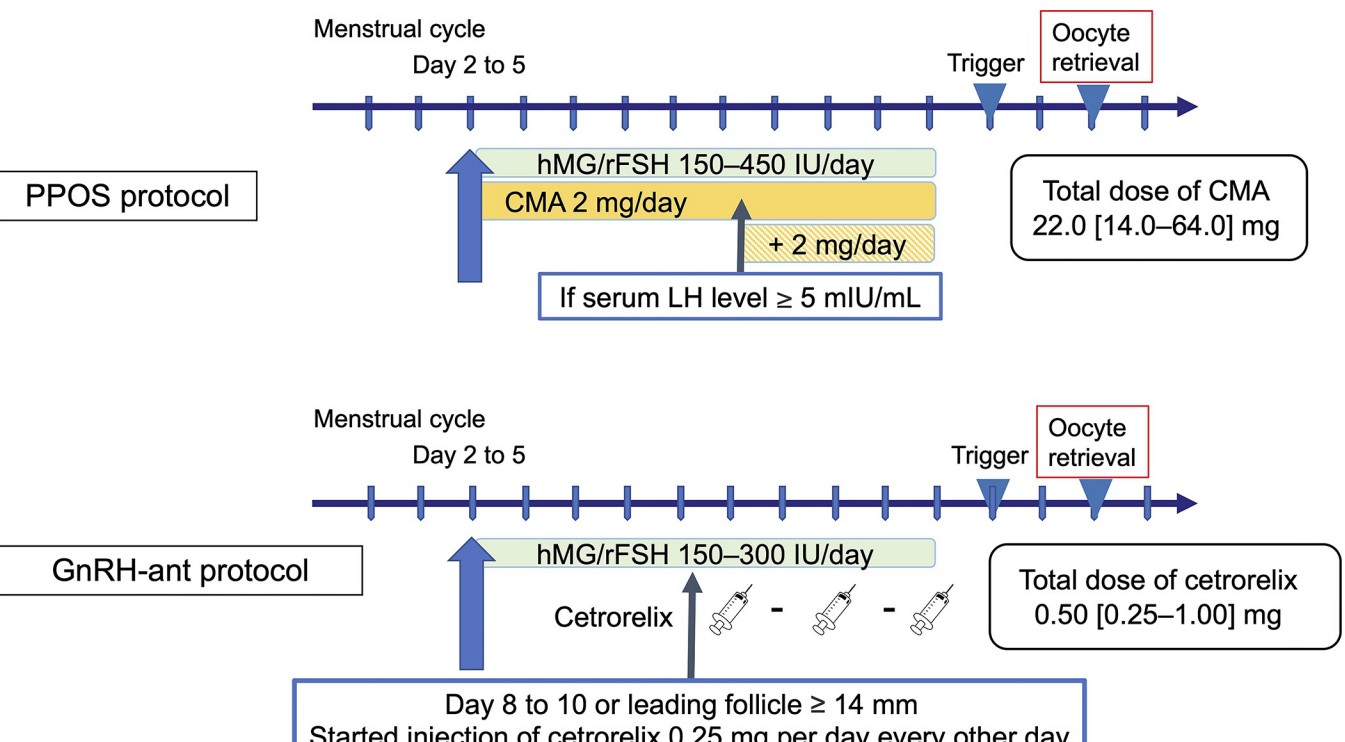

**Fig 2. Controlled ovarian stimulation protocols with the progestin-primed ovarian stimulation (PPOS) and GnRH-antagonist (GnRH-ant) protocols.** This figure outlines the controlled ovarian stimulation (COS) protocols, specifically detailing the methods and schedules of ovulation inhibitor usage for both progestin-primed ovarian stimulation (PPOS) and GnRH-antagonist (GnRH-ant). CMA, chlormadinone acetate; hMG, human menopausal gonadotropin; rFSH, recombinant human follicle-stimulating hormone.

## Ethical approval

This study adhered to the Ethical Guidelines for Medical Research Involving Human Subjects of the Ministry of Health, Labor, and Welfare, Japan. The retrospective design and associated procedures were approved by the Ethical Review Board of Osaka University Hospital (Suita, Osaka, Japan) (No. 19197 and No. 21113–2). In this retrospective study, we consulted the electronic medical records of patients from April 1, 2021 to December 31, 2023. Authors had access to information that could identify individual participants during or after data collection. Samples were collected between November 1, 2021 and February 1, 2022 from patients who provided written informed consent before inclusion and participated in COS cycles at the Reproduction Clinic Osaka, Osaka, Japan.

## Results

### Dose-dependent effects of CMA on oocyte maturation and fertilization rate

The median age of the 299 patients in the PPOS group was 34.0 years (Table 1). The rate of premature LH surge, defined as a serum LH level greater than 10 mIU/m, was 4.7%. The median total dosage of CMA was 22 mg (IQR 18.0–32.0), and 78.9% (236/299) of the patients had a fixed daily CMA dose (Table 2). Univariable Poisson regression analysis revealed a statistically significant dose-dependent increase in the oocyte maturation rate, with an unadjusted RR of 1.004 per 1 mg of CMA (95% CI: 1.001–1.008; $p = 0.017$). The fertilization rate was 1.002 per 1 mg of CMA (95% CI: 0.987–1.017; $p = 0.770$; Table 3, Fig 3). However, after adjusting for

**Table 2. Dosage characteristics of CMA and cetrorelix in patients undergoing the PPOS or GnRH-ant protocols.**

| | PPOS |
|---|---|
| | $n = 299$ |
| **Total dose of CMA (mg) (median, IQR)** | 22.0 [18.0–32.0] |
| **Maintaining a constant daily dose of CMA during OS ($n$ (%))** | 236 (78.9) |
| | **GnRH-ant** |
| | $n = 605$ |
| **Total dose of cetrorelix (mg) (median, IQR)** | 0.50 [0.50–0.50] |
| **The day for initiating cetrorelix of injection (median, IQR)** | 8.0 [8.0–9.0] |

This table demonstrates the specific dosage employed in each stimulation protocol. CMA, chlormadinone acetate; IQR, interquartile range; OS, ovarian stimulation; PPOS, progestin-primed ovarian stimulation

the 17 covariates in a multivariable Poisson regression analysis, the aRR of the maturation rate was 1.003 (95% CI: 0.999–1.007; $p = 0.194$; Table 3). After adjusting for the 18 covariates, the aRR of the fertilization rate was 1.002 (95% CI: 0.985–1.020; $p = 0.783$; Table 3). Multivariable Poisson regression analysis indicated no dose-dependent effects of CMA on oocyte maturation or fertilization rates during the PPOS protocol.

## Dose-dependent effects of cetrorelix on oocyte maturation and fertilization rate

After excluding 3 patients who exhibited signs of ovulation prior oocyte retrieval, 605 patients who underwent GnRH-ant were included. The median age of the 605 participants in the GnRH-ant group was 35.0 years. The rate of premature LH surge was 19.0%. The median total cetrorelix dosage was 0.5 mg [IQR 0.5–0.5], and the median day on which cetrorelix injection was initiated was Day 8 [median, IQR: 8.0–9.0] of the menstrual cycle (Table 2). The univariable Poisson regression analysis revealed no significant differences in the maturation rate, with an unadjusted RR of 1.028 per 0.25 mg of cetrorelix (95% CI: 0.987–1.069; $p = 0.184$), and in the fertilization rate, with an unadjusted RR of 1.038 per 0.25 mg of cetrorelix (95% CI: 0.882–1.221; $p = 0.653$; Table 4, Fig 4). After adjusting for the 17 covariates, the aRR of the maturation rate was 1.009 (95% CI: 0.962–1.059; $p = 0.717$). After adjusting for the 18 covariates, the aRR of the fertilization rate was 1.022 (95% CI: 0.839–1.246; $p = 0.829$). Multivariable Poisson regression analysis revealed no dose-dependent effects of cetrorelix on oocyte maturation or fertilization rates during the GnRH-ant protocol (Table 4).

## Discussion

Our study provides pivotal insights into the dose-dependent effects of CMA and cetrorelix—two widely used ovulation inhibitors—on ART. We analyzed the influence of these inhibitors

**Table 3. Effects of CMA dosage on oocyte maturation and fertilization rates with univariable and multivariable Poisson regression analyses.**

| | Unadjusted | | Adjusted | |
|---|---|---|---|---|
| **Data** | RR [95%CI] | $P$-value | RR [95%CI] | $P$-value |
| **Maturation rate** | 1.004 [1.001–1.008] | **0.017** | 1.003 [0.999–1.007] | 0.194 |
| **Fertilization rate** | 1.002 [0.987–1.017] | 0.770 | 1.002 [0.985–1.020] | 0.783 |

This table outlines the influence of Chlormadinone Acetate (CMA) dosage on oocyte maturation and fertilization rates, examined through both univariable and multivariable Poisson regression analyses. CI, confidence interval; RR, relative risk. Bold text indicates significant $p$-values < 0.05.

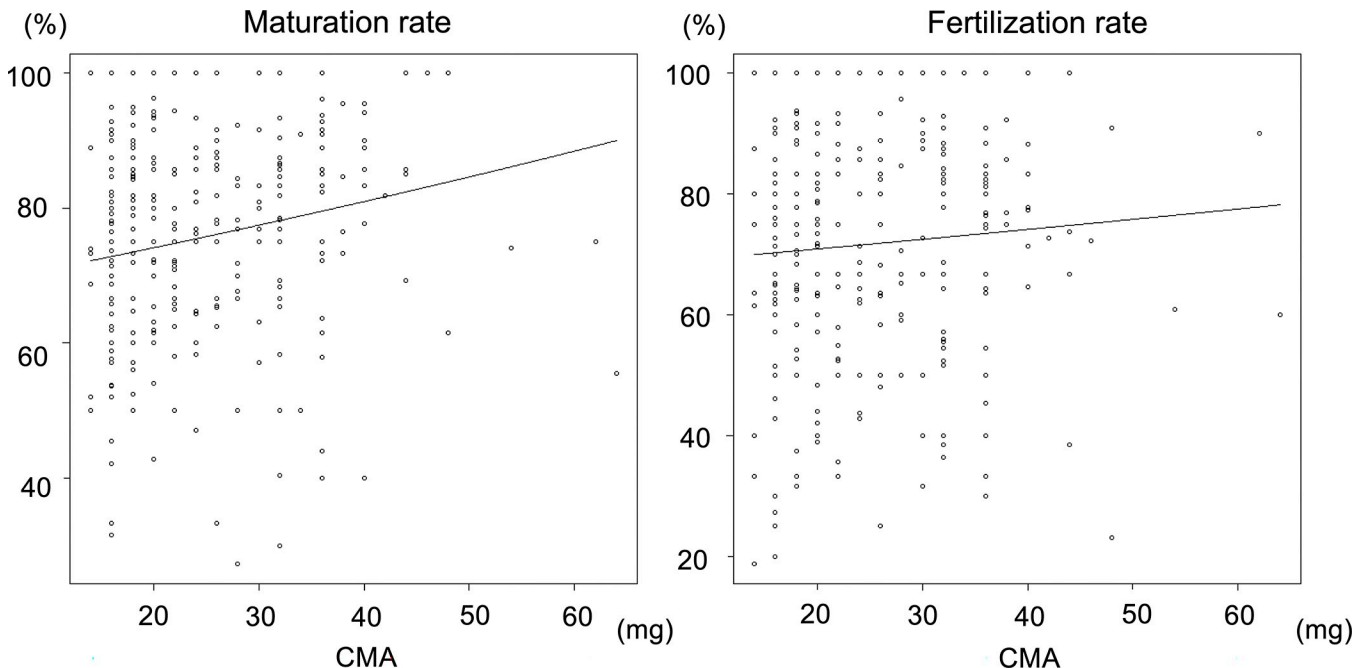

**Fig 3. CMA dosage influence on oocyte maturation rate and fertilization rate with univariable Poisson regression analysis.** This figure illustrates the impact of CMA dosage on oocyte maturation and fertilization rates, analyzed through univariable Poisson regression. CMA, chlormadinone acetate.

on oocyte maturation and fertilization rates in patients with normal ovarian reserve, adjusting for relevant variables. Our findings reveal that within the studied dose range, neither CMA nor cetrorelix significantly affected oocyte quality, suggesting a broad therapeutic window. To our knowledge, this study is the first to demonstrate the association between the total dose of ovulation inhibitors and oocyte maturation and fertilization rate. Importantly, our results underscore the acceptability of administering additional ovulation inhibitors throughout the treatment cycle, if necessary.

The process of ovarian stimulation during ART is critical. It can be broadly divided into three parts: stimulating the ovaries to develop multiple oocytes, preventing premature ovulation of the developed oocytes, and administering a trigger for oocyte maturation [15]. The methods used to prevent ovulation differentiate the ovarian stimulation protocols. Initially, an agonist method was developed using GnRH agonists as ovulation inhibitors, which bind GnRH receptors, leading to receptor desensitization and downregulation, effectively suppressing the LH surge by reducing LH and FSH release from the pituitary gland [5, 16]. This was followed by the popularization of the GnRH-ant method. Unlike agonists, GnRH antagonists directly block GnRH receptors, preventing the release of LH and FSH from the pituitary

**Table 4. Effects of cetrorelix dosage on oocyte maturation and fertilization rates with univariable and multivariable Poisson regression analyses.**

| Data | Unadjusted | | Adjusted | |
|---|---|---|---|---|
| | RR [95%CI] | *p*-value | RR [95%CI] | *p*-value |
| **Maturation rate** | 1.028 [0.987–1.069] | 0.184 | 1.009 [0.962–1.059] | 0.717 |
| **Fertilization rate** | 1.038 [0.882–1.221] | 0.653 | 1.022 [0.839–1.246] | 0.829 |

This table outlines the influence of cetrorelix dosage on oocyte maturation and fertilization rates, examined through both univariable and multivariable Poisson regression analyses. CI, confidence interval; RR, relative risk. Bold text indicates significant *p*-values < 0.05.

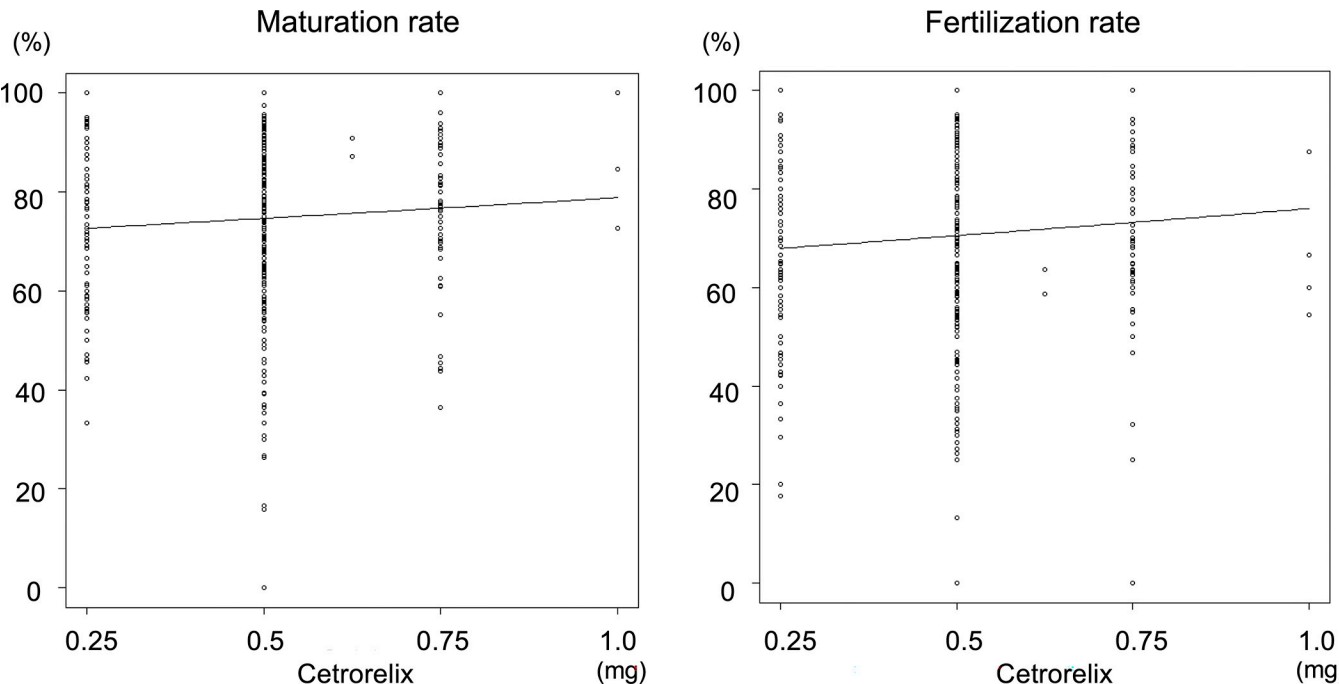

**Fig 4. Cetrorelix dosage influence on oocyte maturation rate and fertilization rate with univariable Poisson regression analysis.** This figure illustrates the impact of cetrorelix dosage on oocyte maturation and fertilization rates, analyzed through univariable Poisson regression.

gland, thus inhibiting ovulation. This rapid onset of action, without the initial flare-up effect associated with agonists, offers a considerable advantage in clinical ART protocols [17]. Recently, the PPOS protocol, which uses progestins as ovulation inhibitors, has gained attention [7]. While these ovulation inhibitors effectively suppress the LH surge, careful adjustment is necessary as LH and FSH are also essential for oocyte development [18]. As the primary factor in follicular development, FSH requires sustained elevation for the proper development of multiple follicles, typically achieved through exogenous FSH administration [19]. Our study suggests that despite suppressing endogenous FSH secretion by ovulation inhibitors, administering adequate amounts of exogenous FSH likely mitigates any dose-dependent adverse effects of these inhibitors on oocyte maturation rates.

The importance of FSH and LH in follicular development is evident in their biological functions and clinical ART outcomes. Indeed, groups receiving only FSH during ovarian stimulation exhibit lower fertilization and pregnancy rates than those administered hMG containing FSH and LH [20]. A meta-analysis further reported enhanced productivity in hMG-administered groups [21]. However, some studies have shown that outcomes with hMG and only FSH are equivalent, suggesting that the addition of LH may not always lead to superior results [22, 23] Despite various studies reporting comparable results between the amount of GnRH antagonist daily administered and ART outcomes [11, 24–27], there is a lack of detailed dose-dependent evaluations. Our study addresses this gap by uniquely analyzing the dose-dependent effects of GnRH antagonists on oocyte maturation and fertilization rates using multivariable Poisson regression analyzes. This detailed dose-response analysis provides valuable insights into the impact of GnRH antagonist dosages on oocyte quality. Importantly, our findings indicated no significant dose-dependent adverse effects of GnRH antagonists on oocyte maturation or fertilization rates. However, lower LH levels at the beginning of stimulation with GnRH-ant protocols have been associated with poorer ART outcomes [28], suggesting that the

initial state of the pituitary gland and ovaries, rather than in-cycle LH levels, plays a crucial role in determining the success of oocyte retrieval.

The use of progestins in PPOS is acknowledged for its role in suppressing LH surges during controlled ovarian stimulation in IVF treatments [7]. Introduced in 2015 [29], PPOS has garnered attention for its straightforward approach and ease of administration, leading to global adoption in various fertility clinics. Despite its apparent benefits in managing LH surges, the precise mechanisms by which PPOS inhibits ovulation are not fully understood [7]. Moreover, while various meta-analyses suggest no significant differences in reproductive outcomes between PPOS and GnRH-ant, the overall efficacy of PPOS, particularly in scenarios excluding fresh embryo transfer [7, 30], continues to be a subject of investigation and debate [31–34]. Several studies have examined the correlation between the amount of progestin used and ART outcomes, with some demonstrating comparable in terms of the number of oocytes retrieved and pregnancy outcomes [10], whereas others report increased mature oocyte rates in the higher-dose group [35]. Our retrospective study demonstrated that within the specific range of progestin usage in our protocol, there was no significant impact on oocyte maturation or fertilization rates in patients with normal ovarian reserve after multivariable Poisson regression analysis, although univariable Poisson regression analysis revealed increased mature oocyte rates in a dose-dependent manner.

The effect of progesterone on oocyte development is complex and not fully understood. Although progesterone receptors A and B are not present in oocytes, progesterone may exert an indirect effect via granulosa cells, potentially through paracrine signaling [36]. Elevated progesterone levels can lead to lower pregnancy rates after fresh embryo transfer owing to endometrial desynchronization [37]. Additionally, high progesterone levels at the time of ovulation triggering may reduce the quantity of high-quality embryos and subsequent pregnancy and birth rates [38–41]. However, this topic remains controversial, with conflicting studies leaving this issue unresolved [42, 43]. Moreover, the specific mechanism by which progesterone inhibits ovulation remains a subject of research [44]. Various progestins are currently being explored in PPOS protocols; however, the specific type and dose required to achieve optimal ART outcomes are yet to be determined [45].

Poisson regression analysis provided greater analytical depth than traditional group comparison methods, enabling an exploration of the continuous spectrum of ovulation inhibitor dosages and their nuanced impacts on oocyte maturation and fertilization rates. By adjusting for multiple covariates, we more accurately isolated the specific effects of CMA and cetrorelix, offering a more comprehensive understanding beyond what simple binary comparison can achieve. The application of Poisson regression analysis is particularly suited for this context, allowing precise modeling of count data and addressing non-linear relationships effectively. This method also manages overdispersion and variability within the data, providing clearer insights into how varying dosages of ovulation inhibitors affect oocyte quality. Furthermore, this statistical approach enables us to draw conclusions about the potential safety and efficacy of different dosing regimens, which could be crucial for tailoring treatments to individual patient profiles in ART. Our findings pave the way for future investigations using similarly sophisticated methodologies to unravel intricate dose–response relationships [46].

This study has certain limitations. First, it was a single-center study limited to Japanese women with normal ovarian reserve, potentially influencing the generalizability of the results. Hence, further studies with diverse populations are required to achieve broader applicability. Additionally, the effects of different types of progestins in similar settings should be explored. Second, as a retrospective study, inherent bias potentially exists. Third, our study did not evaluate pregnancy rates. Although reports suggest that GnRH antagonists may affect endometrial receptivity, we primarily employed a freeze-all strategy, negating potential impacts on the

endometrium. Finally, although our findings showed no adverse effects on oocytes within the tested dose range of ovulation inhibitors for this particular group of patients, this may not be the case for other protocols or patient groups. Nevertheless, the strengths of our study include a larger sample size compared to previous studies comparing dosages and the application of Poisson regression analysis, providing a more comprehensive and accurate analysis than binary comparisons.

## Conclusions

Our study findings indicate that, within the employed dosage ranges, neither CMA nor cetrorelix significantly affects oocyte maturation or fertilization rates in patients with normal ovarian reserve. These findings underscore the suitability of increasing ovulation inhibitor dosages, if necessary. Our results provide valuable guidance to clinicians regarding the strategic utilization of ovulation inhibitors to enhance the effectiveness of infertility treatments.

## Acknowledgments

We express our gratitude to the individuals and embryologists at the Reproduction Clinic Osaka for their crucial contributions to this work. Special appreciation goes to Editage (www. editage.jp) for English language editing.

## Author Contributions

**Conceptualization:** Mika Handa, Tsuyoshi Takiuchi.

**Data curation:** Mika Handa.

**Formal analysis:** Sumika Kawaguchi, Sho Komukai, Tetsuhisa Kitamura.

**Funding acquisition:** Mika Handa.

**Project administration:** Tsuyoshi Takiuchi.

**Resources:** Yasuhiro Ohara, Masakazu Doshida, Takumi Takeuchi, Hidehiko Matsubayashi, Tomomoto Ishikawa.

**Supervision:** Tsuyoshi Takiuchi, Tadashi Kimura.

**Validation:** Sumika Kawaguchi.

**Visualization:** Mika Handa, Sho Komukai, Tetsuhisa Kitamura.

**Writing – original draft:** Mika Handa, Tsuyoshi Takiuchi.

**Writing – review & editing:** Tsuyoshi Takiuchi, Hidehiko Matsubayashi, Tadashi Kimura.

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
