## [Decision Letter · Decision Letter 0]

10 May 2024

PONE-D-24-04395Investigating dosage effects of ovulation Inhibitors on oocyte maturation in assisted reproductive technology: A retrospective study among patients with normal ovarian reservePLOS ONE

Dear Dr. takiuchi,

Thank you for submitting your manuscript to PLOS ONE. After careful consideration, we feel that it has merit but does not fully meet PLOS ONE’s publication criteria as it currently stands. Therefore, we invite you to submit a revised version of the manuscript that addresses the points raised during the review process.

The manuscript and the reviewers’ comments were carefully evaluated. The Reviewers appreciated the manuscript; however, they highlighted some major points that require revisions before considering the manuscript for publication. Suggested improvements and changes are in detail reported in the Reviewers’ comments. 

We look forward to receiving your revised manuscript.

Kind regards,

Simone Garzon

Academic Editor

PLOS ONE

Journal Requirements:

Reviewers' comments:

Reviewer's Responses to Questions

**Comments to the Author**

1. Is the manuscript technically sound, and do the data support the conclusions?

Reviewer #1: Yes

Reviewer #2: Partly

2. Has the statistical analysis been performed appropriately and rigorously? 

Reviewer #1: Yes

Reviewer #2: Yes

3. Have the authors made all data underlying the findings in their manuscript fully available?

Reviewer #1: Yes

Reviewer #2: Yes

4. Is the manuscript presented in an intelligible fashion and written in standard English?

Reviewer #1: No

Reviewer #2: Yes

5. Review Comments to the Author

Reviewer #1: A lot of effort was put into the study by the authors, and it was a large and comprehensive study. But focusing is not easy to understand. Therefore, simplification should be made in both introduction, discussion and tables.

Since there are many studies on GnRH antagonists, it would be appropriate to focus only on the PPOS group.

Although it is reasonable to compare follicular fluid parameters between groups, it is not possible to make a meaningful evaluation with so little data. It should be either removed from the text or the number should be increased if possible.

In Table 5, it is paradoxical that the number of retrieved oocytes is high in the low dose group, although the dose of AMH and gonadotropin used is low.

It was stated that in some patients, additional clomiphene citrate and aromatase inhibitors were given.A comparison between these groups should be given in Table 1.

Reviewer #2: The Ms presented by Mika Handa et. al aims to investigate if there is dose-dependent effect between two LH surge inhibitors and oocyte maturation/fertilization in two corresponding controlled ovarian stimulation protocols used in patients with normal ovarian reserve. The retrospective study uses Poisson regression analyses to establish the dose-dependent effect. The authors claim to conclude that there is no dose-dependent effect between the two LH surge inhibitors and oocyte maturation/ fertilization rate in their used dosage range. The conclusion has some clinical implications regarding the use of LH surge inhibitors, especially the use of progestin.

There are several comments as follows:

1. The aim to use progestin and antagonist in COS is to inhibit premature LH surge. There is no report of the occurrence of premature LH surge in the MS. And any such cases should be excluded in the analyses of dose-dependent effect.

2. Please describe the definition of maturation rate clearly. It is different in IVF and ICSI.

3. As authors stated, HMG may be more productive than r-FSH and, therefore, is a possible influential factor of oocyte maturation. Why not include Gn type in the covariates?

4. The CMA is a less used progestin in PPOS. Why use 2mg per day?

5. Because it seems that a high proportion of subjects (estimated 80% using more than 20mg CMA, and 40% more than 30mg CMA from the figure 3) had a higher total dose of CMA than the cutoff value of 18mg to define high and low dose groups in FF study, the results of no statistical differences of several proteins in FF seems to be of little persuasiveness.

6. Finally, please specify the reproductive outcome in results, conclusion, discussion, and implication to oocyte maturation/fertilization. Oocyte maturation/fertilization are far more from oocyte developmental competent. Up to date, there are still scare data regarding the genetic and epigenetic alteration in the oocytes, and subsequent embryos and offspring from PPOS. Anyway, from this MS, we can only conclude oocyte maturation and fertilization may not be affected by increasing the progestin, such as CMA, if necessary, in COS.

6. PLOS authors have the option to publish the peer review history of their article (what does this mean?). If published, this will include your full peer review and any attached files.

Reviewer #1: No

Reviewer #2: No

---

## [Author Response · Author response to Decision Letter 0]

26 Oct 2024

Reviewer 1

Thank you for the thorough review of our manuscript.

Comment 1:

A lot of effort was put into the study by the authors, and it was a large and comprehensive study. But focusing is not easy to understand. Therefore, simplification should be made in both introduction, discussion and tables. Since there are many studies on GnRH antagonists, it would be appropriate to focus only on the PPOS group.

Response: 

Thank you for your positive feedback regarding the scope and comprehensiveness of our study. We acknowledge that the breadth of the study may have made it difficult to maintain a clear focus, particularly in the introduction, discussion, and tables. In response, we have carefully reviewed these sections and have made significant simplifications to improve clarity and readability. The introduction and discussion now present the key findings more concisely, and we have streamlined the tables to focus on the most relevant data.

As for your suggestion to focus exclusively on the PPOS group, we agree that this could be a potential direction. However, as multivariable Poisson regression analyses on the dose-dependent effects of GnRH antagonists are scarce in the literature, we believe our broader analysis provides important insights into the effects of GnRH antagonist dosages. Additionally, Reviewer 2 did not specifically request focusing solely on the PPOS group, making it difficult to fully comply with this suggestion. Therefore, in lines L319-323, we have refined the sections related to GnRH antagonists to enhance clarity, particularly in the discussion. We hope these revisions balance the scope of our study with a clearer presentation of the results.

Comment 2:

Although it is reasonable to compare follicular fluid parameters between groups, it is not possible to make a meaningful evaluation with so little data. It should be either removed from the text or the number should be increased if possible.

Response:

As per your suggestion, and given the limited sample size for the follicular fluid analysis, we have removed this section from the manuscript. We agree that a meaningful evaluation cannot be conducted with such a small dataset, and thus, to maintain the integrity of the study, we have opted to exclude it entirely.

Comment 3:

In Table 5, it is paradoxical that the number of retrieved oocytes is high in the low dose group, although the dose of AMH and gonadotropin used is low.

Response:

Thank you for your sharp observation regarding the paradoxical relationship between the number of retrieved oocytes and AMH/gonadotropin levels. As you pointed out, there is a potential bias in the data due to two patients in the low dose group who had more than 40 oocytes retrieved, which may have skewed the results. Additionally, the patients in the low dose group were slightly younger, which could have contributed to better ovarian responsiveness. Nevertheless, as you correctly noted, the sample size may not be sufficient to draw definitive conclusions. Therefore, in accordance with your comment and as indicated in Comment 2, we have decided to remove this data from the analysis.

Comment 4:

It was stated that in some patients, additional clomiphene citrate and aromatase inhibitors were given. A comparison between these groups should be given in Table 1.

Response:

Thank you for your valuable suggestion. In response to your comment, we have added a section for "Oral medications administered for ovarian stimulation" in Table 1, and included the relevant data for clomiphene citrate and aromatase inhibitors. This comparison between groups is now clearly presented in the revised table.

Once again, we appreciate your insightful comments and believe that the revisions have significantly strengthened our manuscript. We look forward to your further feedback.

Reviewer 2

We would like to sincerely thank you for the thorough review of our manuscript.

Comment 1:

The aim to use progestin and antagonist in COS is to inhibit premature LH surge. There is no report of the occurrence of premature LH surge in the MS. And any such cases should be excluded in the analyses of dose-dependent effect.

Response: 

Thank you for your insightful comment. As you correctly mentioned, the primary purpose of using ovulation inhibitors in COS is to prevent a premature LH surge. On the other hand, it could also be stated that the ultimate goal of inhibiting premature LH surge is to prevent ovulation before oocyte retrieval through the appropriate use of ovulation inhibitors. There are various definitions of premature LH surge, and some argue that even if a premature LH surge occurs, as long as it does not lead to pre-retrieval ovulation, it may not affect the outcome of oocyte retrieval. In response to your suggestion, we have now explicitly reported the incidence of premature LH surge and ovulation in the manuscript. Furthermore, cases with pre-retrieval ovulation were excluded from the analysis, and we have reanalyzed the data accordingly, specifically in lines L112–117 and Figure 1.

Comment 2:

Please describe the definition of maturation rate clearly. It is different in IVF and ICSI.

Response: 

Thank you for your valuable comment. In response, we have added a detailed description of the definition of maturation rate in the Materials and Methods section (line L181–186). We have clarified the differences in how maturation rate is defined between IVF and ICSI to ensure the distinction is clear.

Comment 3:

As authors stated, HMG may be more productive than r-FSH and, therefore, is a possible influential factor of oocyte maturation. Why not include Gn type in the covariates?

Response: 

Thank you for your insightful comment. As you mentioned, there are reports suggesting that HMG may be more potent than r-FSH, while others suggest they are equivalent (PMID: 32395637, 19356754). In our initial analysis, we did not classify based on Gn type. However, in response to your suggestion, we have now included Gn type as a covariate in the analysis. Specifically, we have categorized the Gn types into three groups: rFSH, rFSH+hMG, and hMG, and reanalyzed the data accordingly. We have also updated Table 1.

Comment 4:

The CMA is a less used progestin in PPOS. Why use 2mg per day?

Response: 

We appreciate your comment. Although the number of reports on this topic is limited, a study by Y. Takeshige et al. (Fertility & Reproduction, Vol. 02, No. 01, pp. 21-26, 2020) demonstrated that CMA at a dose of 2.0 mg/day also has an ovulation suppression effect. Based on this evidence, we selected the 2.0 mg/day dosage for this study.

Comment 5:

Because it seems that a high proportion of subjects (estimated 80% using more than 20mg CMA, and 40% more than 30mg CMA from the figure 3) had a higher total dose of CMA than the cutoff value of 18mg to define high and low dose groups in FF study, the results of no statistical differences of several proteins in FF seems to be of little persuasiveness.

Response: 

Thank you for your valuable comment. As you correctly pointed out, and in line with Reviewer 1’s suggestion to remove the follicular fluid data, we have decided to exclude the analysis of follicular fluid from the manuscript.

Comment 6:

Finally, please specify the reproductive outcome in results, conclusion, discussion, and implication to oocyte maturation/fertilization. Oocyte maturation/fertilization are far more from oocyte developmental competent. Up to date, there are still scare data regarding the genetic and epigenetic alteration in the oocytes, and subsequent embryos and offspring from PPOS. Anyway, from this MS, we can only conclude oocyte maturation and fertilization may not be affected by increasing the progestin, such as CMA, if necessary, in COS.

Response: 

Thank you for your insightful comment. As you correctly pointed out, our study specifically analyzed the effects of ovulation inhibitors on oocyte maturation and fertilization rates. While oocyte maturation and fertilization rates are important factors that could influence reproductive outcomes, it would not be accurate to directly discuss the impact of ovulation inhibitors on reproductive outcomes based solely on this study. Therefore, we have replaced the term "reproductive outcome" with "oocyte maturation/fertilization rate" where applicable. Additionally, when referring to potential impacts on reproductive outcomes, we have been careful to avoid implying a direct link between ovulation inhibitors and reproductive outcomes. This revision allows for a more precise representation of our findings.

---

## [Decision Letter · Decision Letter 1]

22 Dec 2024

Investigating dosage effects of ovulation Inhibitors on oocyte maturation in assisted reproductive technology: A retrospective study among patients with normal ovarian reserve

PONE-D-24-04395R1

Dear Dr. takiuchi,

We’re pleased to inform you that your manuscript has been judged scientifically suitable for publication and will be formally accepted for publication once it meets all outstanding technical requirements.

Kind regards,

Simone Garzon

Academic Editor

PLOS ONE

Additional Editor Comments (optional):

Reviewers' comments:

Reviewer's Responses to Questions

**Comments to the Author**

1. If the authors have adequately addressed your comments raised in a previous round of review and you feel that this manuscript is now acceptable for publication, you may indicate that here to bypass the “Comments to the Author” section, enter your conflict of interest statement in the “Confidential to Editor” section, and submit your "Accept" recommendation.

Reviewer #1: All comments have been addressed

2. Is the manuscript technically sound, and do the data support the conclusions?

Reviewer #1: Yes

3. Has the statistical analysis been performed appropriately and rigorously? 

Reviewer #1: Yes

4. Have the authors made all data underlying the findings in their manuscript fully available?

Reviewer #1: Yes

5. Is the manuscript presented in an intelligible fashion and written in standard English?

Reviewer #1: Yes

6. Review Comments to the Author

Reviewer #1: Thanks for your effort .I have no additional comment .All comments have been addressed well thanks again

7. PLOS authors have the option to publish the peer review history of their article (what does this mean?). If published, this will include your full peer review and any attached files.

Reviewer #1: No

---

## [Editor Report · Acceptance letter]

26 Dec 2024

PONE-D-24-04395R1 

PLOS ONE

Dear Dr. Takiuchi, 

I'm pleased to inform you that your manuscript has been deemed suitable for publication in PLOS ONE. Congratulations! Your manuscript is now being handed over to our production team.

Kind regards, 

on behalf of

Dr. Simone Garzon 

Academic Editor

PLOS ONE